# IFN-γ mediates Paneth cell death via suppression of mTOR

Alessandra Araujo[1], Alexandra Safronova[1], Elise Burger[1],
Américo López-Yglesias[1], Shilpi Giri[1], Ellie T Camanzo[1], Andrew T Martin[1],
Sergei Grivennikov[2,3], Felix Yarovinsky[1]*

[1]Center for Vaccine Biology and Immunology, Department of Microbiology and Immunology, University of Rochester Medical Center, New York, United States; [2]Department of Medicine and Department of Biomedical Sciences, Cedars-Sinai Medical Center, Los Angeles, United States; [3]Cancer Prevention and Control Program, Fox Chase Cancer Center, Philadelphia, United States

**Abstract** Paneth cells constitutively produce antimicrobial peptides and growth factors that allow for intestinal homeostasis, host protection, and intestinal stem cell replication. Paneth cells rely heavily on the glycolytic metabolic program, which is in part controlled by the kinase complex Mechanistic target of rapamycin (mTORC1). Yet, little is known about mTOR importance in Paneth cell integrity under steady-state and inflammatory conditions. Our results demonstrate that IFN-γ, a crucial mediator of the intestinal inflammation, acts directly on murine Paneth cells to alter their mitochondrial integrity and membrane potential, resulting in an TORC1-dependent cell death mechanism distinct from canonical cell death pathways including apoptosis, necroptosis, and pyroptosis. These results were established with the purified cytokine and a physiologically relevant common Th1-inducing human parasite *Toxoplasma gondii*. Given the crucial role for IFN-γ, which is a cytokine frequently associated with the development of inflammatory bowel disease and compromised Paneth cell functions, the identified mechanisms underlying mTORC1-dependent Paneth cell death downstream of IFN-γ may provide promising novel approaches for treating intestinal inflammation.

*For correspondence:
felix_yarovinsky@URMC.
Rochester.edu

## Introduction

Paneth cells, the main antimicrobial peptide producers in the small intestine, constitutively secrete a broad spectrum of bactericidal proteins including defensins, cryptdin-related sequence peptides, and Lysozyme 1 (*Bevins and Salzman, 2011*; *Clevers and Bevins, 2013*; *Sato et al., 2011*; *Adolph et al., 2013*). Beyond regulating the microbiota through the secretion of antimicrobial peptides, Paneth cells play crucial roles in regulating intestinal stem cells during steady-state and stress conditions caused by starvation, aging, tissue damage, or inflammation (*Sato et al., 2011*; *Hodin et al., 2011*; *Pentinmikko et al., 2019*; *Jones et al., 2019*; *Sato et al., 2009*; *Vaishnava et al., 2008*; *Rodríguez-Colman et al., 2017*).

Mechanistic target of rapamycin (mTOR) senses and responds to nutrients, growth factors, and stress (*Saxton and Sabatini, 2017*). Paneth cell-expressed mTOR regulates stem cell self-renewal, as is particularly evident in models of intestinal recovery and caloric restriction (*Yilmaz et al., 2012*; *Igarashi and Guarente, 2016*), emphasizing a non-Paneth cell-autonomous role for mTOR in intestinal homeostasis, whereas Paneth cell-intrinsic roles for mTOR are not well defined. Important modulators of Paneth cell functions and survival are proinflammatory cytokines, especially IFN-g (*Raetz et al., 2013*; *Farin et al., 2014*). We and others have previously observed that IFN-γ is a pleiotropic cytokine that plays a central role in the induction of severe intestinal inflammation characterized by the selective loss of intestinal Paneth cells (*Raetz et al., 2013*; *Farin et al., 2014*; *Burger et al., 2018*).

These observations raise the questions of how IFN-γ mediates Paneth cell loss, whether this cytokine is capable of inducing Paneth cell death, and whether direct or indirect effects of IFN-γ are involved in these processes.

## Results

Cellular responses to IFN-γ are mediated by its heterodimeric cell-surface receptor (IFN-γR), which is widely expressed by epithelial and immune cells (*Bach et al., 1997*; *O'Neill et al., 2016*). To examine whether the direct effects of IFN-γ on Paneth cells are responsible for Paneth cell loss during acute gastrointestinal infection, mice with an intestinal pan-epithelial cell- or Paneth cell-restricted deficiency in IFN-γRII were generated by crossing Vil1-Cre or Defa4IRES-Cre (Paneth cell-specific, PC-Cre) mice, respectively (*Burger et al., 2018*), with Infgr2 $^{flox/flox}$ mice (*Tcyganov et al., 2021*) and orally infected with *Toxoplasma gondii*. While wild-type (WT) animals (Cre-negative Infgr2 $^{flox/flox}$ mice) lost practically all detectable Paneth cells, as was evident in a quantitative analysis of the representative Paneth cell-specific transcripts for the antimicrobial peptides lysozyme 1 (Lyz1), defensin alpha 23 (Defa23), and Defa24 (*Figure 1A*, *Figure 1—source data 1*) and histological quantification of the loss of these cells (*Figure 1B and C*). Epithelial or Paneth cell-restricted IFN-γRII deficiency largely prevented the loss of Paneth cells (*Figure 1A,C*, *Figure 1—source data 1*). An additional analysis of Paneth cells through immunofluorescence-based detection of Lyz1-expressing cells at the base of the intestinal crypts further supported the finding that selective deletion of IFN-γRII in Paneth cells prevented their elimination during *T. gondii* infection (*Figure 1D*, *Figure 1—source data 1*). These results revealed that the direct effects of IFN-γ on Paneth cells are responsible for the rapid loss of these cells during acute microbial infection.

To examine whether Paneth cell death or degranulation results in the apparent disappearance of these cells in response to *T. gondii*-induced IFN-γ, in vivo labeling with propidium iodide (PI), which binds to DNA in dying cells, was next applied. In naïve mice, PI positivity was largely restricted to dying enterocytes located at the top of the villi, and we did not detect PI-positive Paneth cells, while acute infection with the parasite resulted in the selective acquisition of PI positivity by Paneth cells (*Figure 1E and F* and *Figure 1—figure supplement 1A*). A similar conclusion was reached with genetically labeled Paneth cells that allowed integrity analysis in vivo (*Figure 1—figure supplement 1B*). We observed that both membrane and nuclear reporters of Paneth cells lost fluorescence signals during *T. gondii* infection, strongly indicating that the parasitic infection caused the loss of Paneth cells via induction of cell death (*Figure 1—figure supplement 1B* and data not shown).

## IFN-γ-dependent loss of Paneth cells occurs independently of canonical cell death

To dissect the cell death pathway responsible for Paneth cell loss during *T. gondii* infection, we performed experiments in which the role of apoptosis, which is known to be involved in intestinal cell death, was investigated via a terminal deoxynucleotidyl transferase dUTP nick end labeling (TUNEL) assay, and the expression of cleaved Caspase 3, an essential mediator of apoptosis was also evaluated. We observed that Paneth cells did not acquire TUNEL- or cleaved Caspase 3-positivity on any of the examined days post infection, ruling out apoptosis as the major mechanism of Paneth cell death (*Figure 2*), even though it was previously reported that in some rare cases IFN-γ is capable to activate caspase-3/7-dependent pathway of cell death in Paneth cells (*Eriguchi et al., 2018*).

In addition, inhibition of apoptosis by caspase inhibitor zVAD failed to prevent Paneth cell death (*Figure 2—figure supplement 1*). Importantly, zVAD treatment of the intestinal organoids prevented the loss of the stem cell-specific transcript, signifying a role for apoptosis in stem cell death caused by IFN-γ, and suggesting that in contrast to the intestinal stem cells (*Kretzschmar and Clevers, 2019*; *Takashima et al., 2019*), Paneth cells undergo a different mechanism of death in response to this cytokine (*Figure 2—figure supplement 1*). Two additional cell death pathways that are associated with inflammatory responses were next investigated. During pyroptosis, intracellular activation of inflammasomes and Caspase 1 or 11 results in rapid formation of plasma membrane pores and cell lysis (*Broz et al., 2020*; *Place and Kanneganti, 2018*). In agreement with the zVAD treated mice, the absence of Caspases 1 and 11 failed to prevent the loss of Paneth cells, further confirming no role for these caspases in Paneth cell death. The absence of the additional key proteins involved in pyroptosis,

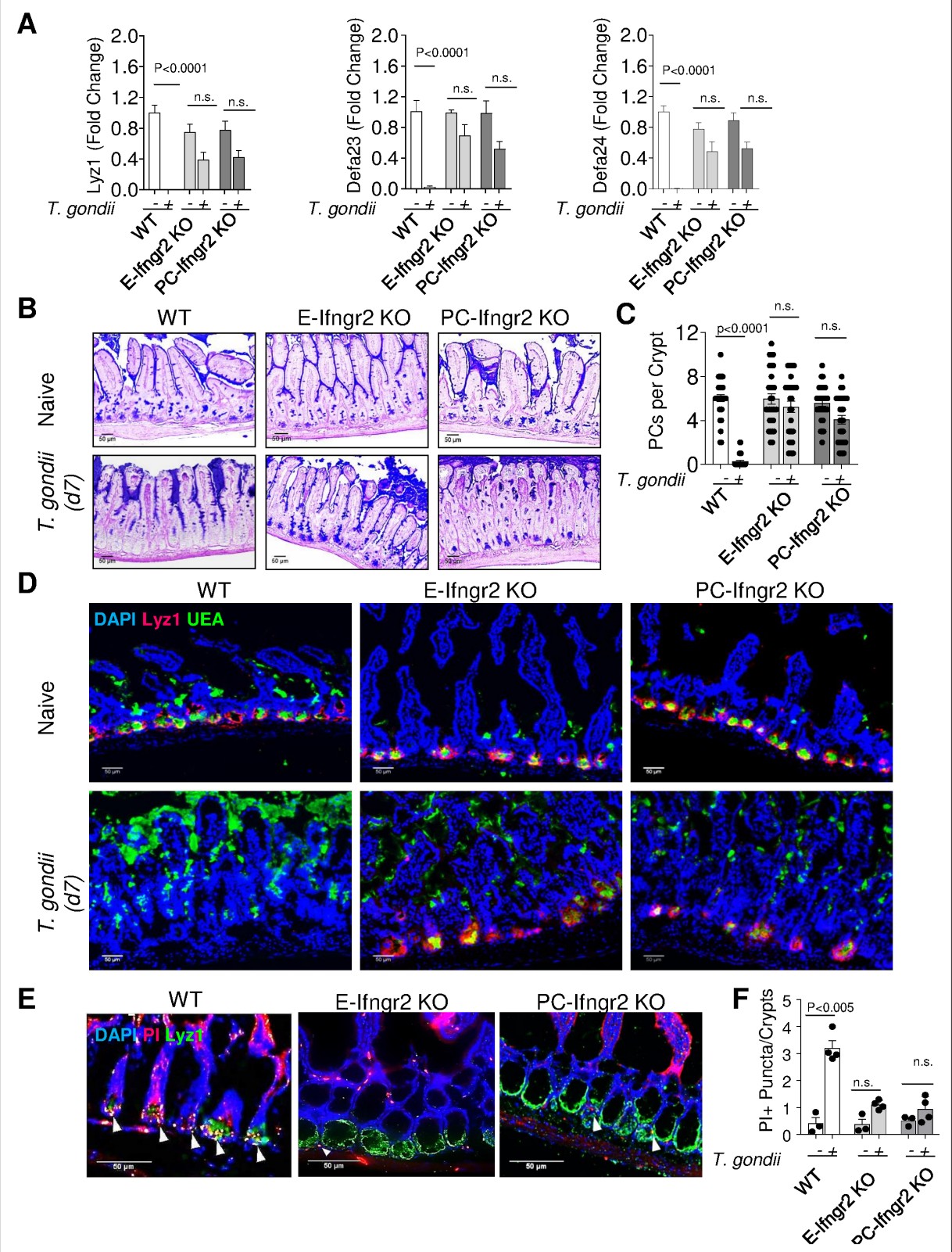

**Figure 1.** IFN- γ directly contributes to Paneth cell loss during *Toxoplasma gondii* infection. (**a**) qRT-PCR analysis of relative Lyz1, Defa23, and Defa24 expression measured in the small intestines of WT, E-Ifngr2 KO, and PC- Ifngr2 KO infected orally with 20 cysts per mouse of the ME49 *T. gondii* on day 7 after infection (n=6) or uninfected controls (n=6). The results are representative of three independent experiments. (**b**) Histological visualization of Paneth cells in the small intestines of naïve and infected WT, E- ifngr2 KO, and PC- ifngr2 KO mice. (**c**) Quantification of Paneth cells per crypt in naïve

*Figure 1 continued on next page*

*Figure 1 continued*

and infected WT, E- Ifngr2 KO, and PC- Ifngr2 KO mice with 20 cysts of ME49 *T. gondii* on day 7. Error bars=mean± s.d. (**d**). Immunofluorescence of intestinal crypts of naive and *T. gondii*-infected WT, E- Ifngr2 KO, and PC- Ifngr2 KO for Paneth cells (UEA, green), Lyz1 (red), and nucleic acid (DAPI, blue). Scale bar: 50 μm. The results are representative of three independent experiments. (**e**) Immunofluorescence images of intestinal crypts of naïve or *T. gondii*-infected WT, E- Ifngr2 KO, and PC- Ifngr2 KO mice treated intravenously with propidium iodide (PI) on day 6 post infection. Arrows show individual PI⁺ puncta. The results are representative of four independent experiments each containing at least three mice per group. Scale bar: 50 μm. (**f**) Average numbers of PI⁺ puncta per crypt. Histological sections shown were selected by blinded observation. The results are representative of four independent experiments, PI (pink), Lyz1 (green), and DAPI (blue). Error bars=mean± s.d.

The online version of this article includes the following figure supplement(s) for figure 1:

**Source data 1.** Analysis of Paneth cells in the small intestines of WT, E-Ifngr2 KO, and PC- Ifngr2 KO infected orally with *Toxoplasma gondii*.

**Figure supplement 1.** Visualization of Paneth cell death during *Toxoplasma gondii* infection in vivo.

including NLRP3 and ASC (encoded by Pycard) (*Orning et al., 2019*), failed to prevent the death of Paneth cells, eliminating pyroptosis as the mediator of Paneth cell death during *T. gondii* infection (*Figure 2—figure supplement 2A and B*).

Similarly, we ruled out a role for necroptosis, a cell death mechanism that depends on RIPK1 and RIPK3, in Paneth cell death downstream of IFN-γ via analysis of mice deficient in RIPK3 (*Figure 2—figure supplement 2C and D*) and animals treated with necrostatin (*Figure 2—figure supplement 2E*), a potent and well-characterized inhibitor of RIPK1 (*Choi et al., 2019*). We observed that inactivation of RIPK1 or RIPK3 had no detectable effect on Paneth cell loss during *T. gondii* infection (*Figure 2—figure supplement 2C–2E*). Taken together, these data revealed that IFN-γ-dependent death in Paneth cells was independent of apoptosis, pyroptosis, and necroptosis (*Figure 2*, *Figure 2—figure supplement 1*, *Figure 2—figure supplement 2*).

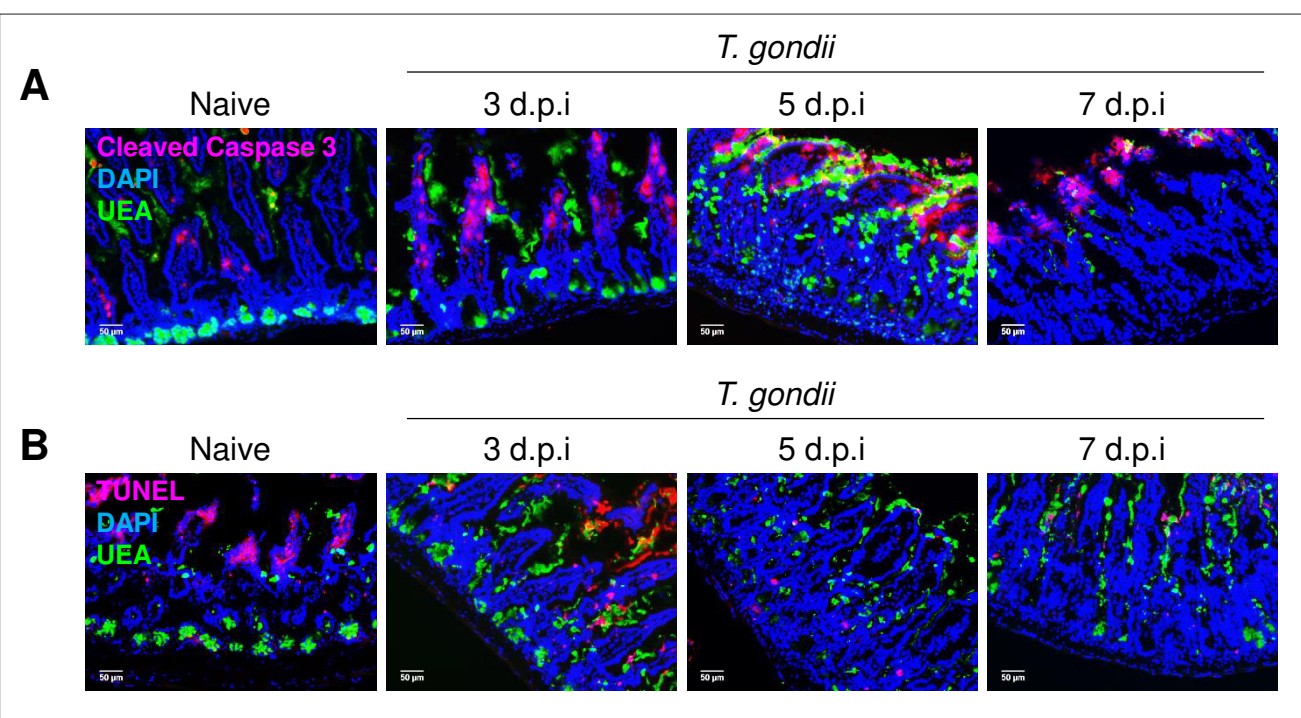

**Figure 2.** IFN-γ triggers Paneth cells death independently of apoptosis. Immunofluorescence images of crypts of WT mice on days 3, 5, and 7 post *Toxoplasma gondii* infection. Images were stained with UEA (green), DAPI (blue), and (**a**) Cleaved Caspase 3 (pink) or (**b**) TUNEL. Scale bar: 50 μm. The results are representative of three independent experiments, involving at least three mice per group. WT, wild-type.

The online version of this article includes the following figure supplement(s) for figure 2:

**Figure supplement 1.** IFN-γ triggers Paneth cells death independently of apoptosis in vitro.

**Figure supplement 2.** IFN-γ-mediated Paneth cell loss occurs independently of pyroptosis or necroptosis.

## *T. gondii* infection leads to IFN-γ-mediated inhibition of mTORC1 activity in Paneth cells

Infections result in cytokine-mediated metabolic reprogramming, which has been primarily studied in immune cell populations (*O'Neill et al., 2016*) . Among several soluble mediators, IFN-γ is capable of triggering profound metabolic changes in multiple cell types, including epithelial cells (*Nava et al., 2010*). One of the key sensors in cell metabolism is the mTORC1 kinase complex, which plays a critical role in integrating metabolism with signal transduction (*Condon and Sabatini, 2019*). In addition, mTORC1 has been strongly implicated in controlling proper Paneth cell functions (*Yilmaz et al., 2012*; *Barron et al., 2017*; *Sampson et al., 2016*). Tracking the activity of mTOR measured by ribosomal S6 protein phosphorylation (pS6$^{Ser\ 240/244}$) in the intestinal crypts revealed high level of mTOR activity in Paneth cells when compared to stem cells in the PC-Cre x TdTomato x Lgr5-GFP double reporter mice that allowed simultaneous detection of these cell types (*Figure 3—figure supplement 1*). These observations prompted us to examine if IFN-γ leads to Paneth cell death via mTORC1 inactivation. We first analyzed the phosphorylation of the ribosomal S6 protein, a well-characterized marker of mTORC1 activity in mice infected with *T. gondii* and treated with anti-IFN-γ therapy throughout infection. We observed a rapid loss of ribosomal S6 protein phosphorylation in Paneth cells as early as day 3 post infection (*Figure 3A*), and this loss preceded the loss of Paneth cells detected by immunofluorescence staining (*Figure 3A*) or quantitative analysis of Paneth cell-specific transcripts (*Figure 3B*, *Figure 3—source data 1* and data not shown). Successful blockade of IFN-γ in vivo with a neutralizing antibody (*Figure 3A and C*) prevented the loss of ribosomal S6 protein phosphorylation in Paneth cells, which were instead retained throughout the course of the experiments (*Figure 3A and B*). These results reveal that IFN-γ plays a role in modulating mTORC1 activity in Paneth cells in vivo. Additionally, Paneth cells lacking IFN-γRII retained ribosomal protein S6 phosphorylation during *T. gondii* infection (*Figure 3D and E*, *Figure 3—source data 1*), further suggesting that direct effects of IFN-γ on Paneth cells result in the loss of mTORC1 activity in this cell type. In addition, we observed a moderate increase in mTORC signaling in anti-IFN-γ treated animals or in mice lacking IFN-γRII in Paneth cells (*Figure 3A*). These observations suggest that basal IFN-γ, most likely driven by the intestinal microbiota (*Benson et al., 2009*), has a negative impact on mTOR activity.

To formally examine whether IFN-γ alone is capable of impacting mTORC1 activity in Paneth cells, naïve mice were injected with recombinant IFN-γ. This treatment resulted in levels of circulating IFN-γ comparable to those seen in *T. gondii*-infected mice (*Figure 4A*, *Figure 4—source data 1*) and triggered the induction of the representative IFN-stimulated genes, including Cxcl10, Ido1, and Irgm3, in the small intestine (*Figure 4C*). We observed that IFN-γ treatment alone triggered Paneth cell disappearance (*Figure 4B*, *Figure 4—source data 1*) and resulted in a rapid loss of S6 ribosomal protein phosphorylation that preceded Paneth cell loss (*Figure 4D*).

Finally, phosphorylation of the mTOR kinase itself was severely compromised in Paneth cells during IFN-γ treatment (*Figure 4E,F*, *Figure 4—source data 1*). Taken together, these experimental approaches reveal that IFN-γ alone is sufficient to elicit profound inhibition of mTORC1 activity in Paneth cells that precedes their loss in vivo.

## Paneth cell preservation is dependent on mTORC1 signaling

We next formally examined the requirements for an intact Paneth cell-intrinsic mTORC1 signaling pathway in the survival of these cells. To achieve this, mice with targeted deletion of the mTORC1 activator Raptor (PC-Raptor KO mice) or the mTOR kinase (PC-Mtor KO mice) in Paneth cells were generated. We observed that a Paneth cell-restricted deficiency in Raptor or mTOR resulted in nearly complete disappearance of these cells in naïve mice (*Figure 5*). This was evident in immunofluorescence analysis of crypts in the small intestine, which showed that Paneth cells were detected in WT mice but not in PC-Mtor or PC-Raptor KO mice (*Figure 5A*). Similarly, unbiased analysis of Paneth cell-specific transcripts revealed that both mTOR and Raptor1 were required for the presence of Paneth cells and functioned in a cell-intrinsic manner for Paneth cell maintenance (*Figure 5B*, *Figure 5—source data 1*). In addition to the genetic models, we observed that treatment of mice with the mTORC1 inhibitor rapamycin resulted in the rapid loss of Paneth cells (*Figure 5C*), which was also evidenced by reduced expression of Paneth cell-specific transcripts (*Figure 5D*, *Figure 5—source data 1*). When combined, our experiments reveal that Paneth cell-intrinsic mTORC1 activity is essential for the presence of Paneth cells.

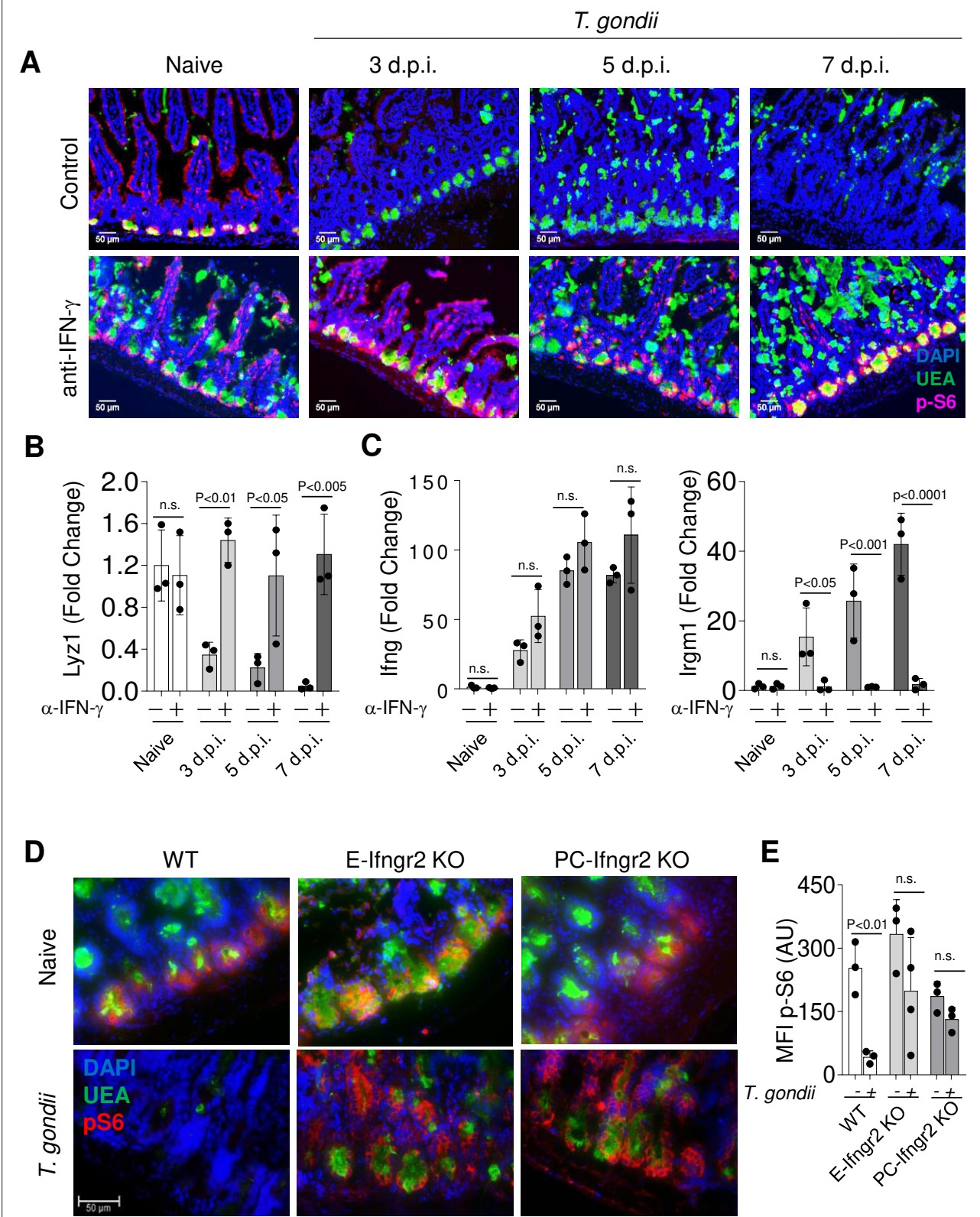

**Figure 3.** *Toxoplasma gondii* infection leads to IFN-γ-mediated inhibition of mTORC1 activity in Paneth cells. (**a**) Immunofluorescence images of small intestinal crypts from naïve, *T. gondii*-infected and *T. gondii*-infected mice treated with anti-IFN-γ antibody on days 3, 5, and 7 post oral infection with *T. gondii*. UEA-positive (green) Paneth cells in crypts were analyzed for p-S6 (p-Ser $^{240/244}$, pink) expression and additionally stained with DAPI (blue). Scale bar: 50 µm. (**b, c**) qRT-PCR analysis for the relative Lyz1 (**b**), IFN-γ, and Irgm1 (**c**) expression in small intestines of naïve and *T. gondii*-infected mice on

*Figure 3 continued on next page*

*Figure 3 continued*

days 3, 5, and 7 post infection. Some mice were additionally treated with IFN-γ neutralizing antibody (α-IFN-g+). The results are representative of three independent experiments, involving three mice per group. Error bars=mean± s.d. (**d**) Immunofluorescence images of small intestinal crypts of naïve and *T. gondii*-infected WT, E-Ifngr2 KO, and PC-Ifngr2 KO mice on day 7 post oral infection with *T. gondii*. UEA positive (green) Paneth cells were stained for p-S6 (p-Ser $^{240/244}$, pink), and DAPI (blue). Scale bar: 50 µm. (**e**) Mean fluorescent intensity (MFI) of p-Ser $^{240/244}$ channel in individual intestinal sections of *T. gondii*-infected WT, E-ifngr2 KO, and PC-ifngr2 KO mice on day 7 post infection. Error bars=mean± s.d.

The online version of this article includes the following figure supplement(s) for figure 3:

**Source data 1.** *Toxoplasma gondii* infection leads to IFN-γ-mediated inhibition of mTORC1 activity in Paneth cells.

**Figure supplement 1.** mTOR activity in Paneth cells and intestinal stem cells in vivo.

## Paneth cell mTORC1 dependency is downstream of *T. gondii* infection-triggered IFN-γ

Our experiments with IFN-γ and rapamycin treatments suggested that during *T. gondii* infection, IFN-γ functions upstream of mTORC1 and causes rapid Paneth cell loss due to its potent inhibitory effect on mTORC1. To formally examine this hypothesis, WT mice were infected with *T. gondii* and treated with rapamycin during the course of infection. As expected, rapamycin treatment prevented the induction of proinflammatory cytokines, including IFN-γ, *in T. gondii*-infected mice due to immunosuppressive functions (*Figure 5—figure supplement 1*). Most importantly, despite completely inhibiting IFN-γ responses in *T. gondii*-infected mice, rapamycin treatment resulted in a loss of Paneth cells that was preceded by a lack of mTOR activity in these cells measured by evaluating the level of the phosphorylated ribosomal S6 protein (*Figure 5—figure supplement 1A*). These results placed inhibition of mTORC1 response downstream of the IFN-γ response in Paneth cells in vivo (*Figure 5—figure supplement 1*).

## IFN-γ-mediated mitochondrial damage leads to mTORC1 inhibition in Paneth cells

The kinase mTOR integrates inputs from growth factor receptors, nutrient availability, and mitochondrial function (*Saxton and Sabatini, 2017*). In an effort to investigate whether mitochondrial alterations contribute to Paneth cell mTORC1 inhibition in response to IFN-γ, we first analyzed mitochondrial integrity in Paneth cells in vivo during *T. gondii* infection. Flow cytometric analysis revealed that by day 5 post infection, when Paneth cell loss is starting to become noticeable (*Raetz et al., 2013*), the remaining Paneth cells while retaining MitoTracker positivity, which allows identification of all mitochondria independent of their membrane potential (*Figure 6A and B*, *Figure 6—source data 1*), were significantly compromised by day 5 post infection in an IFN-γ-dependent manner, as assessed by MitoStatus staining, which detects functionally polarized mitochondria (*Figure 6A,C*). These results were in the full agreement with the previous electron microscopy experiments that revealed extensive damage and vacuolization of the mitochondria in Paneth cells during *T. gondii* infection (*Raetz et al., 2013*).

To further investigate the effects of IFN-γ on mitochondrial functions in Paneth cells, intestinal organoids were generated from WT mice and treated with recombinant IFN-γ. This in vitro system revealed that IFN-γ administration resulted in rapid mitochondrial depolarization, which was followed by the loss of Paneth cells (*Figure 6D, F*). These data suggest that IFN-γ is responsible for the compromised mitochondrial integrity in Paneth cells that leads to mTORC1 inactivation. To formally examine whether the mitochondrial changes downstream of IFN-γ are sufficient to trigger mTORC1 inhibition in Paneth cells, intestinal organoids were treated with FCCP, a potent uncoupler of oxidative phosphorylation in the mitochondria that disrupts ATP synthesis. We established that, similar to IFN-γ and rapamycin treatment, FCCP addition led to a rapid loss of ribosomal S6 phosphorylation (*Figure 6G*). These experiments revealed that disruption of mitochondrial function is sufficient to induce the loss of TORC1 activity and subsequent Paneth cell death (*Figure 6G and H*, *Figure 6—source data 1*). Importantly, rapamycin treatment of intestinal organoids did not result in loss of the inner mitochondrial membrane potential, indicating that mTORC1 functions downstream of mitochondrial functions in Paneth cells (*Figure 6I*).

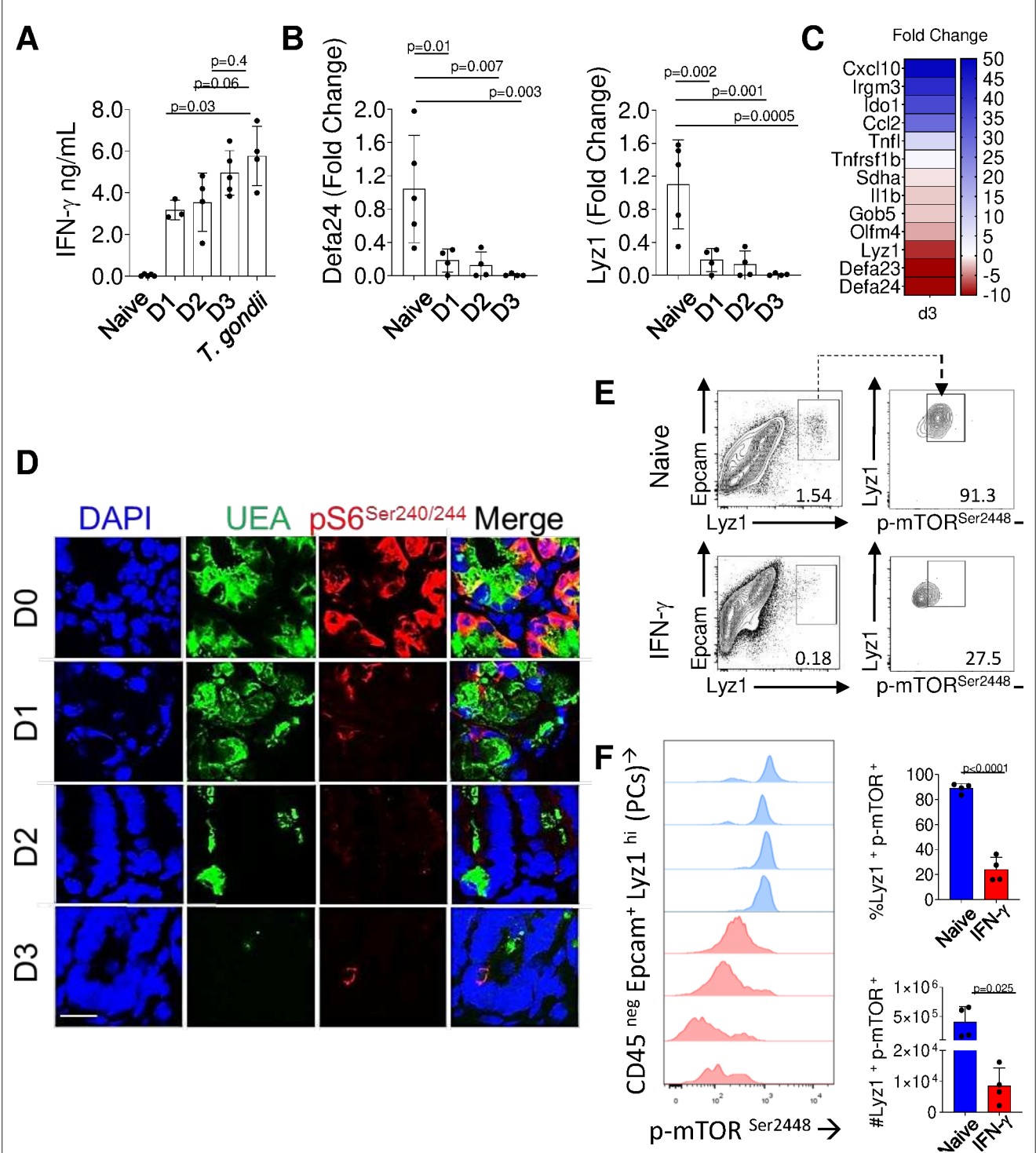

**Figure 4.** IFN-γ is sufficient for inhibition of mTORC1 activity in Paneth cells. (**a**) Mice were treated with recombinant IFN-γ for 24 hr (D1), 48 hr (D2), and 72 hr (D3), and the serum levels of the cytokine were analyzed by ELISA in comparison to *Toxoplasma gondii*-infected mice (day 7post infection). (**b**) qRT-PCR analysis for the relative Defa24 and Lyz1 expression in small intestines of mice treated with recombinant IFN-γ. The results are representative of three independent experiments, each involving at least three mice per group. Error bars=mean± s.d. (**c**) Relative expression of IFN-γ stimulated genes in the small intestines of mice treated with recombinant IFN-γ for 72 hr was analyzed by qRT-PCR. (**d**) Confocal images showing gradual loss of p-S6 Ser240/244 (red) in crypts of mice untreated or treated with recombinant IFN-γ for 24 hr (D1), 48 hr (D2), and 72 hr (D3). DAPI (blue) and UEA (green) staining were used to detect Paneth cells. (**e**) Flow cytometric analysis of phosphorylated mTOR kinase (Ser2448) in Paneth cells (CD45 neg, Lyz1+ Epcam+) from control or mice treated with recombinant IFN-γ for 72 hr. (**f**) The frequency and absolute numbers of mTOR+ Paneth cells in control

*Figure 4 continued on next page*

*Figure 4 continued*

(blue histograms and bars) and IFN-γ treated (red histograms and bars) mice. The results are representative of four independent experiments each containing four mice per experimental group. Error bars=mean± s.d.

The online version of this article includes the following figure supplement(s) for figure 4:

**Source data 1.** IFN-γ is sufficient for inhibition of mTORC1 activity in Paneth cells.

## Intestinal microbiota contributes to mTORC1 dysfunction during acute gastrointestinal infection

The intestinal microbiota has a profound effect on intestinal homeostasis (*Hooper and Macpherson, 2010*; *Garrett et al., 2010*; *Belkaid and Harrison, 2017*). We have previously observed that the presence of the microbiota is required for Paneth cell loss during acute infection with *T. gondii* (*Raetz et al., 2013*), even though the precise understanding of microbiota-mediated effects on Paneth cell death remains incomplete. To examine a potential role for the intestinal microbiota in modulating the IFN-γ-mediated inhibition of mTORC1 during acute *T. gondii* infection, we first analyzed mTOR activity in naïve microbiologically sterile (germ-free [GF]) mice. We observed that naïve germ-free mice exhibited Paneth cell mTOR activity similar to that in conventional (CONV) mice, as was evident from comparable staining for ribosomal protein S6 phosphorylation in the small intestine between CONV and GF mice (*Figure 7*, *Figure 7—source data 1*). Thus, the presence of the microbiota is not required for basal mTOR activity in Paneth cells. In contrast, while parasitic infection resulted in the rapid loss of Paneth cell mTOR activity in CONV mice, GF mice retained phosphorylated ribosomal protein S6 at all examined time points post infection (*Figure 7B and C*, and data not shown), strongly suggesting that the presence of the intestinal microbiota is required for the loss of mTOR activity in Paneth cells. As a result, Paneth cells were largely retained in GF mice, and only a mild loss of Paneth cells was observed in the GF mice compared with CONV mice (*Figure 7*).

To test if microbiota-dependent loss of Paneth cells was due to the contribution of the intestinal bacteria to the potent IFN-γ responses caused by the parasitic infection (*Figure 7—figure supplement 1A*), GF mice were injected with the recombinant IFN-γ and Paneth cells were next analyzed in those mice. We observed that GF mice treated with rIFN-γ (*Figure 7—figure supplement 1B*) lost Paneth cell specific transcripts similarly to CV mice (*Figure 7—figure supplement 1C*), ruling out an intrinsic resistance of Paneth cells to IFN-γ in the absence of the intestinal bacteria. Thus, the reduced levels of IFN-γ seen in GF mice infected with *T. gondii* (*Figure 7—figure supplement 1A*) are responsible for the survival of Paneth cells in the absence of microbiota. Altogether, these data show that the microbiota is required for IFN-γ-mediated mTORC1 dysfunction and the subsequent death of Paneth cells during acute gastrointestinal infection.

## Discussion

IFN-γ is a pleotropic cytokine that is induced in response to all groups of pathogens including viruses, bacteria, and protozoan parasites (*Schroder et al., 2004*; *Shtrichman and Samuel, 2001*; *Taylor et al., 2004*). Much attention has been focused on IFN-γ induced host defense mechanisms; particularly in myeloid cells (*Taylor et al., 2004*; *Shenoy et al., 2007*; *Yamamoto et al., 2012*; *Fabri et al., 2011*; *Boehm et al., 1997*; *Barrat et al., 2019*). The major effects of IFN-γ include, but are not limited to, enhanced antigen processing and presentation (*Matsuo et al., 2004*; *Büning et al., 2005*; *Gerosa et al., 2002*), induction of antimicrobial peptides, and expression of immunity-related GTPases (IRGs) (*Taylor et al., 2004*; *Boehm et al., 1998*), and changes in immunometabolism that leads to the degradation of amino acids required by pathogens for intracellular growth (*Yamamoto et al., 2012*; *Beatty et al., 1994*; *Dai et al., 1994*). Surprisingly and despite an early realization that IFN-γ is also among the central mediators of the immunopathological responses (*Panitch et al., 1987*; *Belnoue et al., 2008*; *Reinhardt et al., 2015*), especially at the mucosal tissues (*Obermeier et al., 1999*; *Heimesaat et al., 2006*), relatively little is known regarding the pathological mechanisms mediated by this cytokine.

We and others have previously reported that IFN-γ is responsible for severe intestinal inflammation characterized by the rapid loss of Paneth cells, a highly specialized type of secretory epithelial cell located in the small intestine that produce large quantities of antimicrobial peptides including

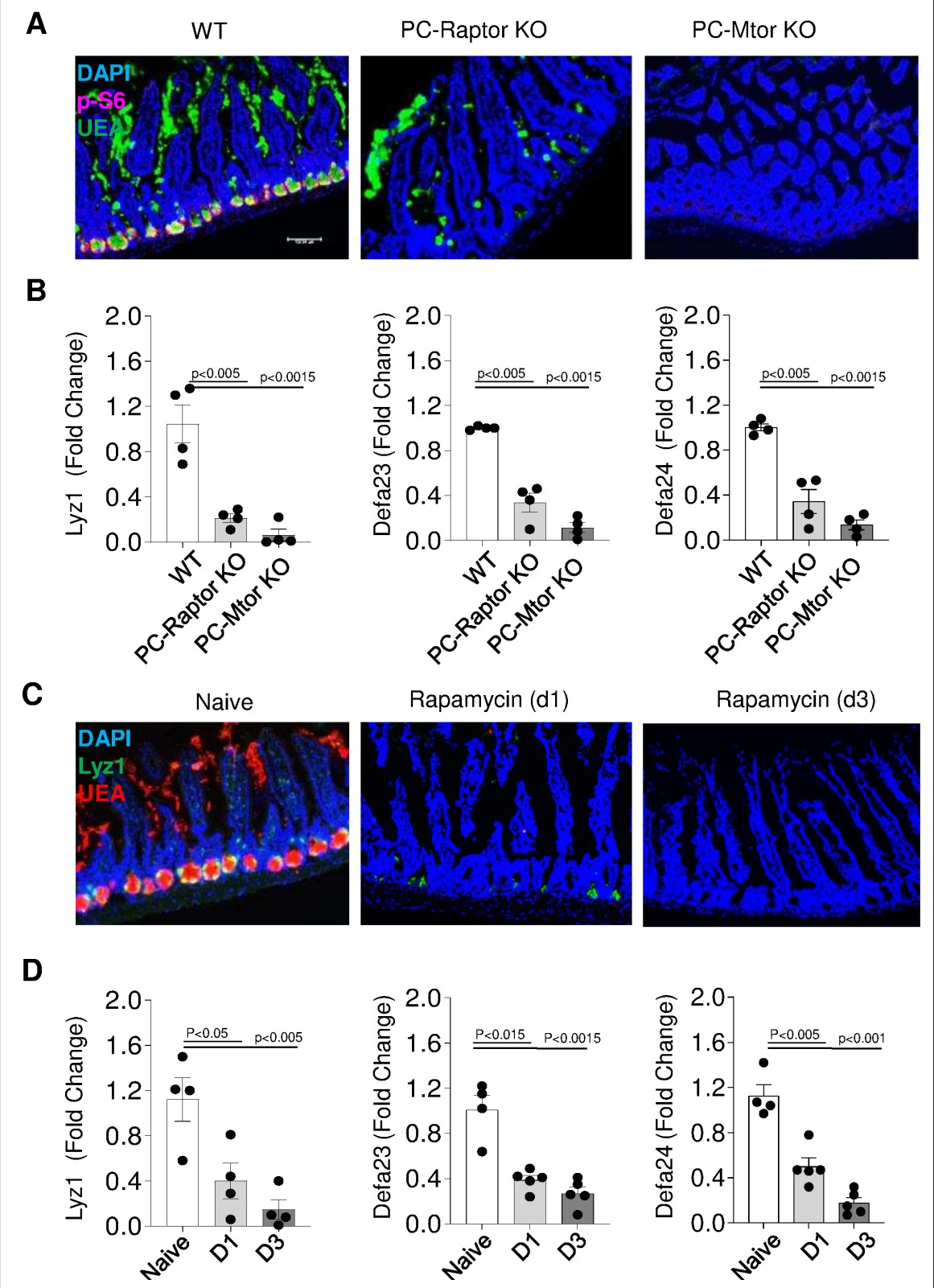

**Figure 5.** Paneth cells require intrinsic mTORC1 signaling. (**a**) Immunofluorescence images of small intestinal crypts of WT mice and mice lacking the mTOR regulatory protein Raptor (PC-Raptor) or mTOR kinase (PC-Mtor) in Paneth cells that were detected by UEA staining (green) and additionally stained with anti-p-S6 antibody (red). DAPI (blue) was used for counterstaining. Scale bars=100 μM. (**b**) qRT-PCR analysis of Lyz1, Defa23, and Defa24 relative expression in the small intestines of WT mice and mice lacking Raptor or mTOR in Paneth cells. Error bars=mean± s.d. (**c**) Immunofluorescence

*Figure 5 continued on next page*

*Figure 5 continued*

images of small intestinal crypts in WT mice treated with rapamycin daily for 24 hr (D1) or 72 hr (D3). Paneth cells were detected by staining for Lyz1 (green), UEA (red), and DAPI (blue). (**d**) qRT-PCR analysis for relative Lyz1, Defa23,and Defa24 expression in the small intestines in WT mice treated with rapamycin daily for 24 hr (**D1**) or 72 hr (D3). Error bars=mean± s.d. The results are representative of three independent experiments. WT, wild-type.

The online version of this article includes the following figure supplement(s) for figure 5:

**Source data 1.** Paneth cells require intrinsic mTORC1 signaling.

**Figure supplement 1.** Paneth cell mTORC1 dependency is downstream of *Toxoplasma gondii*-infection triggered IFN-$\gamma$.

α-defensins and lysozyme (*Raetz et al., 2013*; *Farin et al., 2014*). Paneth cell-derived antimicrobial peptides regulate the composition of the microbiota and are critical for host defense against enteric pathogens (; *Clevers and Bevins, 2013*; *Vaishnava et al., 2008*). While we and others were able to demonstrate that IFN-γ is required for the loss of Paneth cells, whether the direct or indirect effects of this cytokine on Paneth cells causes their disappearance remained unknown. Furthermore, the precise knowledge of the mechanism leading to IFN-γ-mediated Paneth cell death is unknown. This is a significant gap as both experimental and clinical data strongly suggest a link between high IFN-γ levels, impaired Paneth cell function, and inflammatory intestinal disorders (*Fais et al., 1991*; *Matsuoka et al., 2004*; *Krausgruber et al., 2016*). Furthermore, a crucial role for IFN-γ in driving Paneth cell-dependent intestinal inflammation has been established in the mouse models with compromised homeostasis functions controlled by Paneth cell-restricted autophagy (*Burger et al., 2018*; *Grizotte-Lake and Vaishnava, 2018*; *Matsuzawa-Ishimoto et al., 2017*). Direct gene targeting in Paneth cells revealed that compromised cell-intrinsic autophagic machinery enhanced the susceptibility of Paneth cells to proinflammatory cytokines and caused an exaggerated or even lethal intestinal pathology (*Burger et al., 2018*). Thus, in order to understand the mechanism linking IFN-γ to intestinal inflammation, it is essential to understand how IFN-γ triggers the loss of Paneth cells.

In this work, our in vivo experimental data established a mTORC1-dependent mechanism of Paneth cell death in direct response to IFN-γ. The current work revealed that IFN-γ alone was sufficient for mitochondrial damage in Paneth cells. Although apoptosis, necrosis, and pyroptosis can be triggered by mitochondrial damage (*Green, 2019*), these mechanisms of cell death were not detected in Paneth cells. Furthermore, the possible roles for pyroptosis and necroptosis in IFN-γ-mediated Paneth cell death were ruled out by definitive mouse models deficient in those cell death pathways. Instead, we uncovered that a loss of mTORC1 activity downstream of IFN-γ signaling was responsible for Paneth cell death in vivo and in vitro. The results from our work revealed that impaired mitochondrial function caused by IFN-γ is at least in part responsible for mTOR inhibition. This process alone is sufficient to inhibit mTOR in Paneth cells, but cannot exclude other IFN-γ induced pathways that may play a role in mTOR inhibition.

Mouse genetic models with Paneth cell-specific targeting of mTOR and Raptor formally confirmed that intact mTORC1 holds an essential role in the presence of Paneth cells. Overall, our results here established that the direct effects of IFN-γ on Paneth cells caused impaired mitochondrial respiration in Paneth cells that was responsible for mTORC1 inactivation. Given the crucial role for IFN-γ, which is a cytokine frequently associated with the development of inflammatory bowel disease and compromised Paneth cell functions, the identified mechanisms of mTORC1-dependent Paneth cell death downstream of IFN-γ may provide promising novel approaches for treating intestinal inflammation.

## Materials and methods

### Mice

C57BL/6 mice were originally purchased from the Jackson Laboratories and maintained in the pathogen-free animal facility at the University of Rochester of Medicine and Dentistry. Ifngr2$^{flox/flox}$ mice were generated using targeted embryonic stem cells obtained from the Knockout Mouse Project repository and injected into C57Bl/6 albino blastocysts by the Fox Chase Cancer Center Transgenic Mouse Facility as previously described (*Tcyganov et al., 2021*).

Rptor$^{fl/fl}$ (B6.Cg-Rptortm1.1Dmsa/J) and Mtor $^{fl/fl}$ (B6.129S4-Mtortm1.2Koz/J) mice were purchased from Jackson Laboratories and also crossed to Defa4 IRES-Cre mice (*Burger et al., 2018*) to generate PC-Raptor or PC-Mtor conditional knockouts. Mice for all experiments were age- and sex-matched.

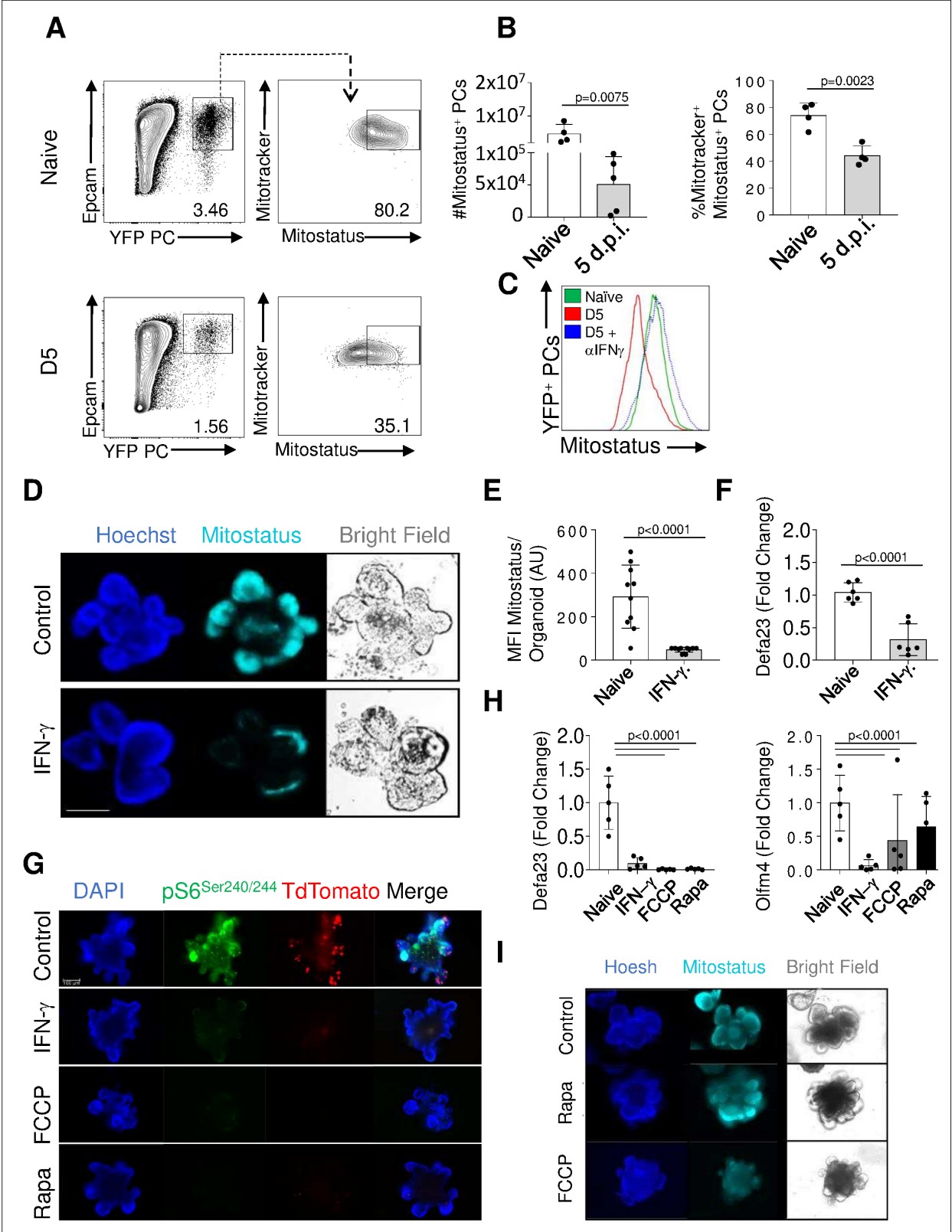

**Figure 6.** IFN-$\gamma$-mediated mitochondrial damage accompanies mTORC1 inhibition in Paneth cells. (**a, b**) Flow cytometric analysis of Paneth cell mitochondrial activity (potentiation) and mass analyzed by mitostatus and mitotracker staining, respectively, in naïve or *Toxoplasma gondii*-infected PC-Cre x Rosa-YFP reporter mice (day 5 post infection). The results are representative of three independent experiments, each involving 3–4 mice per group. Error bars=mean± s.d. (**c**). MitoStatus mean fluorescence intensity (MFI) in the Paneth cells in naïve (green), *T. gondii*-infected (red), and *T.*

*Figure 6 continued on next page*

*Figure 6 continued*

*gondii*-infected and treated with anti-IFN-γ antibody (blue). The results are representative of three independent experiments. (**d**) Intestinal organoids were generated from small intestinal stem cells isolated from crypts of WT mice and stimulated with IFN-γ (200 ng/ml) in vitro for 24 hr and stained with MitoStatus (cyan) and Hoechst (blue). The images shown are representative of 30 organoids imaged across three technical replicate wells per condition. Scale bar: 100 µM. (**e**). MitoStatus MFI of the individual intestinal organoids treated with IFN-γ in vitro for 24 hr and imaged at 40×. (**f**) qRT-PCR analysis for relative Defa23 expression in organoids stimulated with IFN-γ (200 ng/ml) for 24 hr. The results are representative of three independent experiments. (**g**) Intestinal organoids were generated from PC-Cre x TdTomato reporter mice and stimulated with IFN-γ (200 ng/ml), Rapamycin (500 nM) or FCCP (100 µM) for 6 hr (n=4 per treatment). (**h**) qRT-PCR analysis for relative Defa23 and Olfm4 expression in PC-Cre x TdTomato organoids stimulated with IFN-γ (200 ng/ml), Rapamycin (500 nM) or FCCP (100 µM) for 6 hr (n=4 per treatment). Data shown are mean±s.d. (**i**) Intestinal organoids were generated from WT mice and stimulated with Rapamycin (500 nM) for 2 hr. Organoids were stained with Hoescht (blue) and MitoStatus (cyan) prior imaging. The organoid images are representative of 30 organoids imaged across three technical replicates per condition.

The online version of this article includes the following figure supplement(s) for figure 6:

**Source data 1.** IFN-γ-mediated mitochondrial damage accompanies mTORC1 inhibition in Paneth cells.

This study included both male and female mice, and the data derived from male and female mice identified no sex-specific differences in the performed experiments. GF C5BL/6 mice were bred and maintained at the University of Rochester School of Medicine and Dentistry. All animal experimentation in this study was reviewed and approved by the University of Rochester's University Committee on Animal Resources (UCAR), the Institutional Animal Care and Use Committee (IACUC).

## *T. gondii* infection and histopathology

Me49 strain *T. gondii* tissue cyst (bradyzoite) stages were maintained through serial passage in Swiss Webster mice. For infections, brains of chronically infected mice were mechanically homogenized by passage through a series of 18-gauge, 20-gauge, and 22-gauge needles. Experimental mice were orally infected with 20 T. gondii brain cysts (ME49 strain). Portions of small intestine were fixed in Carnoy's fixative, embedded in paraffin, and stained with Alcian blue/Periodic Acid Schiff (PAS). Paneth cells were identified based on their morphology of large granule-containing cells and a baso-lateral nucleus at the base of the intestinal crypt. Quantification of Paneth cells per intestinal crypt was performed in a double blinded manner.

For immunofluorescence images, small intestines were divided into 16 equal parts from duodenum to distal illeum and segment 8 was flushed and flash-frozen in OCT compound (Tissue Tek). 8 µm intestine sections were fixed with Foxp3/Transcription Factor Staining Buffer Set (eBioscience, Cat# 00-5521-00) for 15 min, permeabilized with 0.25% Triton-X in phosphate-buffered saline (PBS), and blocked with 0.1% Triton-X containing 5% Normal Goat Serum in PBS. Sections were stained with primary antibodies against Lysozyme 1 (rabbit polyclonal; Dako) or Phospho-S6 Ribosomal Protein (Ser235/236) (Rabbit mAb#4858; Cell Signaling Technology) in 0.1% Triton-X/PBS with 5% Normal Goat Serum overnight at 4°C. Slides were washed and incubated with donkey anti-rabbit Alexa 568 (Molecular Probes), goat anti-rabbit Alexa 488 (Molecular Probes), and UEA for 1 hr in PBS at room temperature. Slides were mounted in ProLong Gold (Molecular Probes). Specimens were imaged with a Leica SPE system (Leica DMi8) fitted with a Leica 63× objective NA 1.4. or in an inverted Olympus IX81 60×/1.42 Oil objective.

For electron microscopy analysis, the small intestines were fixed in 2.5% glutaraldehyde in 0.1 M sodium cacodylate, followed by 1% osmium tetroxide in 0.1 M sodium cacodylate. The samples were embedded in epoxy resin (University of Rochester Electron Microscopy Core) and polymer-ized at 70°C. Ultrathin sections were cut at 70 nm and stained with uranyl acetate and lead citrate. Sections were examined at 120 kV with a Hitachi 7650 Analytical TEM with an Erlangshen 11 MP digital camera and Gatan software for imaging and morphometric analysis.

In some experiments, mice were injected i.p. with 200 µg anti-IFN-γ (clone XMG1.2, BioXCell) on days 0, 2, 4, and 6 post *T. gondii* infection or 50 µg of recombinant mouse IFN-γalone every 8 hr for 3 consecutive days. In some experiments, mice were intravenously injected with PI on day 6 post *T. gondii* infection for 20 min prior analysis. Quantification of PI positive puncta was performed in a double-blinded manner. For rapamycin experiments, rapamycin (Sigma-Aldrich) was adminis-tered orally daily for a max of 7 days at 7.5 mg/kg in 5% PEG-400 / 5% Tween-80 and reconstituted in 100% EtOH at 10 mg/ml. Necrostatin was administered at (4.5 mg/kg) IP twice daily during *T. gondii* infection.

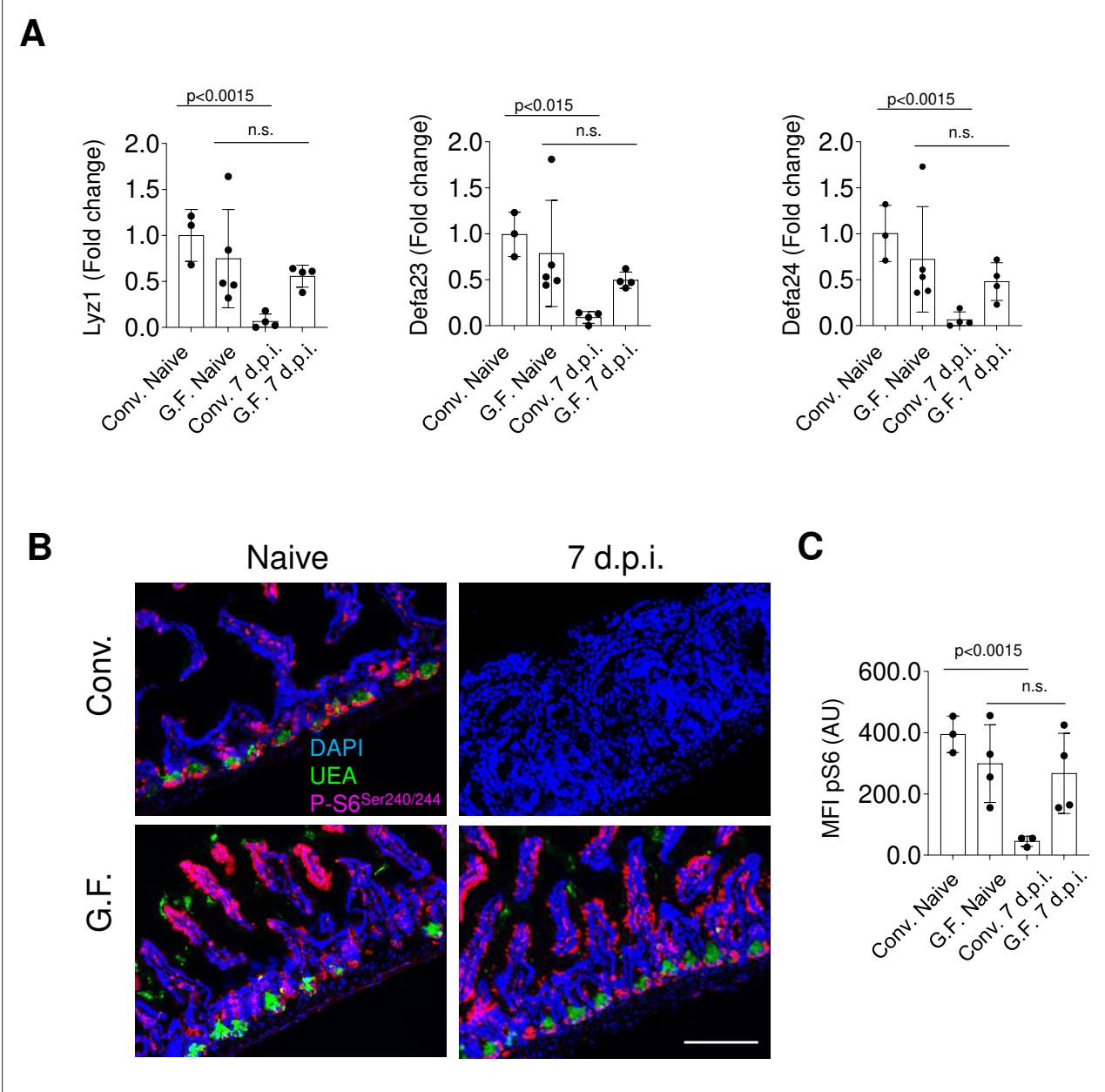

**Figure 7.** Microbiota is required for IFN-$\gamma$-mediated mTORC1 inhibition during *Toxoplasma gondii* infection. (**a**) qRT-PCR analysis for relative expression of Lyz1, Defa23, and Defa24 measured in the small intestines of conventional (CONV) and germ-free (GF) C57Bl/6 mice infected orally with 20 cysts of the ME49 *T. gondii* (day 7 post infection). The results are representative of three independent experiments. Error bars=mean± s.d. (**b**) Immunofluorescence images of small intestinal crypts of naïve and *T. gondii*-infected WT CONV and GF free mice 7 days post oral infection with 20 Me49 *T. gondii* cysts. UEA (green), p-S6 (p-Ser $^{240/244}$) (pink), and DAPI (blue). Scale bar: 200 µM. (**c**) MFI quantification of p-Ser$^{240/244}$ channel in individual intestinal sections of *T. gondii*-infected WT CONV and GF mice on day 7 post infection. Error bars=mean± s.d. The results shown are representative of three separate experiments. Error bars=mean± s.d. MFI, mean fluorescence intensity; WT, wild-type.

The online version of this article includes the following figure supplement(s) for figure 7:

**Source data 1.** Microbiota is required for IFN-$\gamma$-mediated mTORC1 inhibition during *Toxoplasma gondii* infection.

**Figure supplement 1.** Microbiota contributes to IFN-$\gamma$ response during *Toxoplasma gondii* infection.

## Organoid stimulation and imaging

Intestinal organoids were generated from the small intestine as previously described (*Sato et al., 2009*). Briefly, the proximal small intestine was dissected, rinsed with PBS, minced, and incubated in 2.5 mM EDTA in PBS for 30 min at 4°C with gentle agitation. This mixture was passed through a 70 mm cell strainer and pelleted. Isolated crypts were resuspended in Matrigel and plated in 50 µl drops on pre-warmed wells in a 24-well plate. After polymerization, 500 ml of complete growth medium (Advanced DMEM/F-12 supplemented with EGF [Peprotech], Noggin [Peprotech], N2 [Life Technologies], B27 [Life Technologies], Glutamax [Life Technologies], Penicillin/Streptomycin [Life Technologies], HEPES [Life Technologies], N-Acetylcysteine [Sigma-Aldrich], and R-Spondin) was added and refreshed every 2–3 days. Organoids were stimulated by addition of recombinant murine IFN-γ (200 ng/ml; R&D Systems), Carbonyl cyanide 4-(trifluoromethoxy)phenylhydrazone (50 µM; Sigma=Aldrich), or rapamycin (1 µM; Sigma-Aldrich) for 2, 6, or 24 hr. For immunofluorescence images, organoids were either imaged live after the addition of Hoesch (Sigma-Aldrich) and Mitostatus red (BD Biosciences) or underwent intracellular staining after brief fixation with 4% PFA and permeabilization with 0.25% Triton-X. For intracellular staining, samples were incubated with primary antibody against Phospho-S6 Ribosomal Protein (Ser235/236) (Rabbit mAb#4858; Cell Signaling Technology) overnight at 4°C, washed, and incubated for 1 hr with goat anti-rabbit Alexa 488 (Molecular Probes) and DAPI (Molecular Probes). Images of crypt organoids were taken using a Leica DMi8 using the long range 40× objective.

## Quantitative real-time PCR

Total RNA was isolated from the small intestines or organoid well using TRIzol (Life Technologies) and subjected to first-strand cDNA synthesis using iScript Reverse Transcription Supermix for RT-qPCR (Bio-Rad). Real-time PCR was performed using Ssofast EvaGreen Supermix (Bio-Rad) as previously described. The relative expression of each sample was determined after normalization to housekeeping gene HPRT using the ddCt method. The following primers were used for analysis of the gene expression: IFN-γ 5'-ACTGGCAAAAGGATGGTGAC-3' and 5'-TGAGCTCATTGAATGCTTGG-3'; TNF 5'-GCCTCTTCTCATTCCTGCTTGT-3' and 5'- GGCCATTTGGGAACTTCTCAT-3'; TNFR2 5'-CAGG TTGTCTTGACACCCTAC-3' and 5'-GCACAGCACATCTGAGCCT-3'; CXCL10 5'- GACGGTCCGCT-GCAACTG-3' and 5'- GCTTCCCTATGGCCCTCATT-3'; IRGM1 5'- AGCCAACGAGTCCTTGAAGA-3' and 5'-GCACATGTCATCAGCCTCAG-3'; IRGM3 5'- CTGGAGGCAGCTGTCAGCTCCGAG-3' and 5'-GTCCTTTAGAGCTTTCCTCAGGGAGGTCTTG-3'; CXCL10 5'- GACGGTCCGCTGCAACTG-3' and 5'-GCTTCCCTATGGCCCTCATT-3'; IDO1 5'-GTGGGCAGCTTTTCAACTTC-3' and 5'- GGGCTTTGCTC-TACCACATC-3', CCL2 5'- CAGGTCCCTGTCATGCTTCT-3' and 5'- GAGTGGGGCGTTAACTGCA-3'; SDHA 5'- GCTCCTGCCTCTGTGGTTGA-3' and 5'-AGCAACACCGATGAGCCTG-3'; and IL-1b 5'- GCCCATCCTCTGTGACTCAT-3' and 5'- AGGCCACAGGTATTTTGTCG-3'.

Paneth cell transcripts were additionally validated with the following primers that allowed for the selective analysis of Defa23 (5'-TCTGGTATGCTATTGTAGAAC-3' and 5'-GACAGCAGAGCGTGTATA-3') and Defa24 (5'-GATCTGGTATGCTATTGTAGAG-3' and 5'-GACAGCAGAGCATGTACAA-3') as recently described (*Castillo et al., 2019*).

## Flow cytometric analysis

Small intestine segments were washed and incubated in HBSS with 30 mM EDTA for 20 min. Epithelial cells were dissociated mechanically through vigorous shaking followed by incubation in Triple-X (Thermo Fisher Scientific) supplemented with Y-27632 (10 mM, Sigma-Aldrich) and 0.5 mM N-acetyl-L-cysteine at 37°C. Dissociated enterocytes were filtered and pelleted before re-suspended in complete advanced DMEM/F12 media containing 10 mM Y-27632 and 0.5 mM N-acetylcysteine. Cells were stained with fluorochrome-conjugated surface antibodies according to the manufacturer's instructions. For intracellular staining, cells were fixed and permeabilized for 1 hr at 4°C with the Foxp3/Transcription Factor Staining Buffer Set according to the manufacturer's instructions (eBioscience). The following antibodies were used for surface or intracellular staining: CD45 BUV393 (BD Cat#565967), Epcam BV421 (BioLegend, Cat# 324219), p-S6 (S235/236) PerCP e710 (eBioscience, Cat# 46900742), p-mTOR eFluor 660 9S24480 (eBioscience, Cat# 50971842), CD4 BV786 (BD, Cat#563727), IFN-γ BV421 (BD, 563376), Lysozyme 1 (rabbit polyclonal; Dako), and goat anti-rabbit Alexa 488 (Molecular

Probes). Cell fluorescence was measured using an LSRII flow cytometer, and data were analyzed using FlowJo software (Tree Star).

## ELISA analysis

IFN-γ, TNF, CCL2, and IL-12p40 concentrations in the sera were analyzed by standard sandwich ELISA Kit according to manufacturer's instructions (eBioscience).

## Statistics

All data were analyzed with Prism (version 6; GraphPad). These data were considered statistically significant when p-values were less than 0.05 by two-tailed t-test or two-way ANOVA using Tukey's multiple comparisons tests or one-tailed t-tests. Image data sets were processed using Leica Advanced Fluorescence software (Leica) and ImageJ software (National Institutes of Health).

## Acknowledgements

The authors are grateful to Drs. Patrick Cervantes and Laura J Knoll (University of Wisconsin – Madison, Madison, WI) for help with the RIPK3 deficient mice. This work was supported by NCI/NIH R01CA227629 and CA21813 to SIG, and the National Institute of Allergy and Infectious Diseases grants R01AI136538 and R01AI121090, and by the Burroughs Wellcome Foundation to FY.

## Additional information

### Funding

| Funder | Grant reference number | Author |
|---|---|---|
| National Institutes of Health | R01AI136538 | Felix Yarovinsky |
| National Institutes of Health | R01AI121090 | Felix Yarovinsky |
| National Institutes of Health | CA218133 | Sergei Grivennikov |
| National Institutes of Health | CA227629 | Sergei Grivennikov |

The funders had no role in study design, data collection and interpretation, or the decision to submit the work for publication.

### Author contributions

Alessandra Araujo, Conceptualization, Data curation, Formal analysis, Investigation, Methodology, Validation, Writing - original draft, Writing - review and editing; Alexandra Safronova, Conceptualization, Data curation, Investigation, Methodology, Software, Validation; Elise Burger, Data curation, Formal analysis, Investigation, Methodology, Validation; Américo López-Yglesias, Ellie T Camanzo, Andrew T Martin, Methodology; Shilpi Giri, Dr. Giri contributed to Fig 7 of the revised manuscript. All authors agreed with Dr. Giri inclusion and place in the author list, Investigation, Methodology; Sergei Grivennikov, Methodology, Resources; Felix Yarovinsky, Conceptualization, Data curation, Formal analysis, Funding acquisition, Investigation, Project administration, Resources, Supervision, Writing - original draft, Writing - review and editing

### Author ORCIDs

Alessandra Araujo http://orcid.org/0000-0001-7568-074X
Américo López-Yglesias http://orcid.org/0000-0001-6797-2179
Felix Yarovinsky http://orcid.org/0000-0001-5825-8002

### Ethics

All mice were maintained at in the pathogen-free American Association of Laboratory Animal Care-accredited animal facility at the University of Rochester Medical Center, Rochester, NY. All animal

experimentation (animal protocol #102122) has been reviewed and approved by the University Committee on Animal Resources (UCAR).

### Decision letter and Author response

Decision letter https://doi.org/10.7554/eLife.60478.sa1
Author response https://doi.org/10.7554/eLife.60478.sa2

## Additional files

### Supplementary files

• Transparent reporting form

### Data availability

All data generated or analysed during this study are included in the manuscript and supporting files.

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
