## [Decision Letter]

**Acceptance summary:**

Araujo and colleagues report on the role of IFNγ in the inhibition of mTOR pathway and induction of Paneth cell death. The authors use multiple gene deficient mouse strains, in combination with Toxoplasma infection, to systematically define the Paneth-cell autonomous role of IFNγ and mTOR. The investigators also make use of a series of KOs for genes encoding key components of distinct cell death modalities, such as necroptosis and pyroptosis to indicate that none of the known pathways o cell death are involved in this novel finding. The genetic models are complemented with pharmacological approaches, such as inhibitors of cell death pathways or mTOR signaling and the use of intestinal organoids to further probe the mechanism leading to Paneth cell death. Taken together the data clearly demonstrate a role for IFNγ in promoting the death of intestinal Paneth cells via a mTOR dependent pathway.

**Decision letter after peer review:**

Thank you for submitting your article "IFN-γ mediates Paneth cell death via suppression of mTOR" for consideration by *eLife*. Your article has been reviewed by 4 peer reviewers, including Nicola L Harris as the Reviewing Editor and Reviewer #1, and the evaluation has been overseen by Wendy Garrett as the Senior Editor. The following individuals involved in review of your submission have agreed to reveal their identity: Gillian Coakley (Reviewer #2); Carla V Rothlin (Reviewer #4).

The reviewers have discussed the reviews with one another and the Reviewing Editor has drafted this decision to help you prepare a revised submission.

Summary:

Paneth cells produce antimicrobials and growth factors that protect and promote the small intestinal stem cell niche. The authors previously demonstrated that IFN-g produced during *Toxoplasma gondii* infection mediates loss of Paneth cells, a key event that affects intestinal barrier function in this and other disease models. In this study, the authors take advantage of a previously characterized aDefensin4-IRES-Cre mouse to make Paneth cell-specific KOs to investigate cell-intrinsic mechanisms. They demonstrate that IFN-g acts directly on Paneth cells during *T. gondii* infection and induces an atypical cell death. IFN-g was found to mediate this effect through mitochondrial damage that leads to mTOR inhibition. They also demonstrate that loss of mTOR activity in Paneth cells is dependent on the microbiota. Thus, by using *T. gondii* infection as a model to study immunopathology, the authors identify a new Paneth cell-intrinsic role for mTOR in preventing a type of cytokine-induced cell death.

The above results provide important insight into the toxic effect of IFN-g. The genetic techniques clearly establish mTOR within Paneth cells as an important target for this cytokine that has been implicated in many inflammatory settings.

Essential revisions:

A) Regarding the reporting of a novel method of cell death, named "interferonoptosis",

All reviewers agreed that the study falls short from identifying the type of cell death modality involved and that this is a major weakness. While the knowledge that neither apoptosis, necroptosis or pyroptosis appear to account for this type of death is relevant, the results included in this manuscript are not sufficient to coin a new type of cell death (tentatively named as interferonoptosis by the authors). The mechanism involved in inducing the Paneth cell death should be explored in more detail (see notes on further experimental work as outlined below). Alternatively, the authors may choose to perform a subset of these experiments in addition to substantiality 'toning down' their language' regarding the cell death pathway involved and providing a clear discussion of what conclusions can and cannot be drawn from the existing dataset.

– A further confirmation of the ability of IFNγ to directly signal to paneth cells and activated mTORC1 signaling should be provided by adding IFNγ to organoids from the reporter mice using the same assay as described in SFIg4.

– To substantiate their findings that IFN-mediated Paneth cell death is caspase-3 independent, it would be important to address the published findings by Eriguchi et al., (2017) – 10.1172/jci.insight.121886, which reports that "IFN-γ induces Paneth cell death through a caspase-3/7-dependent pathway", again using an enteroid-based system.

– Do the authors know whether they successfully inhibited RIPK1 following administration of necrostatin in Figure S3E? This is potentially of great interest because RIPK1 lies upstream of both apoptosis and necroptosis, and thus, this experiment provides additional support for their conclusion that the cell death modality represents something other than these two forms, which can interconvert. (If technically challenging in vivo, the authors may consider using the organoids to demonstrate a lack of a role for RIPK1).

– What leads to the inhibition of mTOR downstream of IFN γ receptor? This should be tested as multiple mechanisms can lead to inhibition of mTOR. For example, is this event downstream of AMPK, amino acid starvation? Does TSC have a role?

– The assessment of the pathway cannot be limited to measurements p-S6Ser240/244 and should include other markers such as 4E-BP1 phosphorylation. p-S6Ser240/244 staining appears inconsistent across panels. Please see Figure 4A and Figure S6B.

– Given the role of mTOR, mitochondrial damage and the vacuolization detected by EM, is it possible that Paneth cells undergo a type of autophagy-dependent cell death. This is particularly intriguing given the prior report from the same authors on the induction of autophagy by IFNγ in Paneth cells (Burger et al., CHM 2018). The relationship between these events should be explored. Is this dependent on ULK1, ATG13? Is mitophagy selectively being triggered? Ultrastructural changes are important in defining cell death and figure 5C lacks the inclusion of a non-infected control.

– How does IFN γ induced cell death compares to the cell death triggered by other cytokines, such as TNF?

B) Regarding the impact of the microbiota on mTORC and Paneth cells:

The authors also show that intestinal microbiota contributes to mTOR dysfunction and Paneth cell death. However the data shown in Figure S6D is not called out in the results and this should be corrected. Is the reason microbiota depletion rescues mTOR activity because the bacteria are necessary for IFN-g production during *T. gondii* infection? Please clarify this point in the manuscript. It should be noted that degranulation of Paneth cells is not stimulated in germ-free mice. Thus, Paneth cells might be more resilient to mTOR inhibition in germ-free mice. A more quantitative approach for the assessment of mTOR signaling in SPF versus WT mice across time, including earlier time points than 7 days post infection (see Figure S6) and additional markers of mTOR signaling is needed.

---

## [Author Response]

Essential revisions:A) Regarding the reporting of a novel method of cell death, named "interferonoptosis",All reviewers agreed that the study falls short from identifying the type of cell death modality involved and that this is a major weakness. While the knowledge that neither apoptosis, necroptosis or pyroptosis appear to account for this type of death is relevant, the results included in this manuscript are not sufficient to coin a new type of cell death (tentatively named as interferonoptosis by the authors). The mechanism involved in inducing the Paneth cell death should be explored in more detail (see notes on further experimental work as outlined below). Alternatively, the authors may choose to perform a subset of these experiments in addition to substantiality 'toning down' their language' regarding the cell death pathway involved and providing a clear discussion of what conclusions can and cannot be drawn from the existing dataset.

We fully agree with this criticism. We have now avoided the usage of ‘*interferonoptosis’* entirely and instead use precise terminology such as “Paneth cell preservation is dependent on mTORC1 signaling”.

We also performed a subset of the requested experiments for clarification of several important issues raised by the referees in combination with the suggested ‘toning down’ of the claim regarding a novel mechanism of cell death.

– A further confirmation of the ability of IFNγ to directly signal to paneth cells and activated mTORC1 signaling should be provided by adding IFNγ to organoids from the reporter mice using the same assay as described in SFIg4.

The requested experiment was included in our original submission (Figure 5h, now Figure 6g).

As expected, IFN-γ largely eliminated Paneth cells and terminated mTOR signaling indirectly measured by the analysis of P-S6^Ser240/244^ (Figure 6g)

– To substantiate their findings that IFN-mediated Paneth cell death is caspase-3 independent, it would be important to address the published findings by Eriguchi et al., (2017) – 10.1172/jci.insight.121886, which reports that "IFN-γ induces Paneth cell death through a caspase-3/7-dependent pathway", again using an enteroid-based system.

We would like to thank the reviewers and editors for this question. We now discuss this mechanism in our submission. Specifically, we state that*:*

“We observed that Paneth cells did not acquire TUNEL- or cleaved Caspase 3-positivity on any of the examined days post infection, ruling out apoptosis as the major mechanism of Paneth cell death (Figure 2), even though it was previously reported that in some rare cases IFN-γ is capable to activate caspase-3/7-dependent pathway of cell death in Paneth cells (19).”

– Do the authors know whether they successfully inhibited RIPK1 following administration of necrostatin in Figure S3E? This is potentially of great interest because RIPK1 lies upstream of both apoptosis and necroptosis, and thus, this experiment provides additional support for their conclusion that the cell death modality represents something other than these two forms, which can interconvert. (If technically challenging in vivo, the authors may consider using the organoids to demonstrate a lack of a role for RIPK1).

We would like to thank all the reviewers for this very valuable comment. We are confident that RIPK1 was successfully blocked based on the following experiments:

1. We have adopted a protocol for in vivo administration of necrostatin from Immunity 35, 908–918, December 23, 2011 (RIP Kinase-Dependent Necrosis Drives Lethal Systemic Inflammatory Response Syndrome). In agreement with the previous publication (Figure 5), necrostatin administration in vivo completely prevented TNF-induced mortality in mice. We have not included those technical experiments as the purpose of those experiments was to validate the commercial reagents and establish an experimental protocol in the lab.

While it is theoretically possible that we failed to completely inhibit RIPK1 in Paneth cells, we hope the reviewers would agree that even partial RIPK1 inhibition should prevent some Paneth cell death if this pathway plays a significant role in IFN-γ mediated Paneth cell death. However, we observed no effect of necrostatin on Paneth cell death.

2. We have attempted to analyze RIPK1 in sort-purified Paneth cells in response to *T. gondii* infection (on day 6 post infection prior a complete loss of Paneth cells). We have included the data for the reviewers’ benefit in Author response image 1. As evident from two independent experiments, while a potent STAT-1 phosphorylation was observed in sort-purified Paneth cells in response to *T. gondii* infection, we could not detect a phosphorylated form of RIPK1 in those cells. Please note that while this may look like a trivial experiment, it required multiple mice to purified sufficient numbers of Paneth cells (200,000) from *T. gondii* infected mice on day 6 post infection due to their substantial loss. We fully acknowledge that WB is a crude approach to analyze the kinase activity of RIPK1, and we hope that these additional data satisfies the reviewers’ technical concern as they revealed that Paneth cells responded to IFN-γ (phospho-STAT-1), but did not demonstrate potent or at least easily detectable RIPK1 activation.

**Author response image 1. sa2fig1:** Western blot analysis of sort-sort-purified Paneth cells isolated from naïve or *T. gondii* infected mice (day 6 post infection). Paneth cells were combined from 3 naïve mice (I) and 12-15 *T. gondii* infected mice (T). The data shown are from two independent experiments.

– What leads to the inhibition of mTOR downstream of IFN γ receptor? This should be tested as multiple mechanisms can lead to inhibition of mTOR. For example, is this event downstream of AMPK, amino acid starvation? Does TSC have a role?

The results from our manuscript revealed that impaired mitochondrial function caused by IFN-γ is at least in part responsible for mTOR inhibition. This process alone is sufficient to inhibit mTOR in Paneth cells (now Figure 6G). We also fully acknowledge that other pathways may play a role in mTOR inhibition. Specifically, we state that:

“The results from our work revealed that impaired mitochondrial function caused by IFN-γ is at least in part responsible for mTOR inhibition. This process alone is sufficient to inhibit mTOR in Paneth cells, but cannot exclude other IFN-γ induced pathways that may play a role in mTOR inhibition”

We have analyzed several other possible pathways, especially IFN-γ induced amino acid starvation by analyzing iNOS knockout mice or by blocking Trp metabolism, but those experiments revealed that IFN-γ induced iNOS played no role in Paneth cell loss. We have not included those negative data as they would further dilute the main message of the manuscript (IFN-γ mediated mitochondrial dysfunction is responsible for mTOR inactivation and Paneth cell death).

We appreciate the suggestion regarding TSC as the TSC complex may mediate nutrient signaling upstream of mTOR to regulate Paneth cell functions and survival. We are in the process of generating Paneth cell specific TSC KO mice using Tsc2 flox mice (Jax #027458). These mice were cryopreserved, but we took the recommendation seriously and acquired those mice as we anticipate a significant Paneth cell dysfunction in the absence Tsc2 as TSC1/2 antagonizes the mTOR signaling pathway. We hope nevertheless that the reviewers would agree that this analysis is beyond the scope of this submission and may reveal *‘gain of function’* phenotype in Paneth cells.

– The assessment of the pathway cannot be limited to measurements p-S6Ser240/244 and should include other markers such as 4E-BP1 phosphorylation. p-S6Ser240/244 staining appears inconsistent across panels. Please see Figure 4A and Figure S6B.

We do not conclude that the mTOR pathway is involved in Paneth cells death based on the measurements of p-S6Ser^240/244^. Instead, the major evidence comes from the genetic data (now Figure 5A) that formally confirmed that the mTOR pathway is essential for Paneth cells. We strongly believe that the genetic data fully support our claims. We would like to stress that p-S6Ser^240/244^ is a common readout for mTOR activity and the measurements of p-S6Ser^240/244^ were perform to analyze mTOR activity without any statement regarding the downstream events associated with this particular mTOR function in Paneth cells. In addition, we also analyzed phospho-mTOR Ser^2448^ (Figure 4e, 4f).

We respectfully disagree that there is an inconsistency across the panels shown in Figure 4A (now Figure 5A) and Figure S6B (now Figure 7B). Both panels revealed high intensity for p-S6 protein in Paneth cells detected in both conventional (CV) and germ-free (GF) mice. The staining pattern of p-S6 protein in epithelial cells is distinct between GF and CV mice; however, this difference is significant in that the variation is not an experimental inconsistency, but most likely due to a physiological response to microbiota. We hope the reviewers would agree that the analysis of microbiota-dependent mTOR activation in enterocytes (not Paneth cells) is beyond the scope of this submission, even though it is an interesting research direction.

If the reviewer pointed that there were some variations for UEA staining, we would like to clarify that UEA detects both Paneth cells and mucus-producing Goblet cells. We see no variations for the detection of Paneth cells via UEA staining. As for the mucus detection, it is fully expected that mucus staining varies across the mice as it is very sensitive to the fixation protocol.

– Given the role of mTOR, mitochondrial damage and the vacuolization detected by EM, is it possible that Paneth cells undergo a type of autophagy-dependent cell death. This is particularly intriguing given the prior report from the same authors on the induction of autophagy by IFNγ in Paneth cells (Burger et al., CHM 2018). The relationship between these events should be explored. Is this dependent on ULK1, ATG13? Is mitophagy selectively being triggered? Ultrastructural changes are important in defining cell death and figure 5C lacks the inclusion of a non-infected control.

We have previously extensively tested this hypothesis (*Burger et al., CHM 2018*) and in our previous publication we have specifically state that:

“We initially hypothesized that additional induction of IFN-γ during the mucosal response to T. gondii leads to excessive autophagy activation and unintentional Paneth cell elimination. Contrary to our hypothesis, the experimental analysis of T. gondii-infected PC-Atg5 KO and E-Atg5 KO mice revealed that elimination of Atg5 only in Paneth cells or in all enterocytes did not protect Paneth cells from IFN-γ-dependent cell death. Instead, we revealed that the lack of Atg5 resulted in profound intestinal immunopathology characterized by near complete elimination of all epithelial cells, including Paneth cells.”

We believe the previous results from my lab (*Burger et al., CHM 2018*) and an independent study from Kevin Maloy (Cell Host and Microbe, Volume 23, Issue 2, 14 February 2018) along with the detailed analysis of the above mentioned publications by Mayara Grizotte-Lake and Shipra Vaishnava (CHM, 2018) have already largely addressed the raised question.

We initially included the electron microscopy experiments (previous Figure 5C) as an additional illustration for the new data shown in Figure 5 (now Figure 6). As the detailed ultrastructural changes in Paneth cells during *T. gondii* infection has been previously characterized by our lab (Nature Immunology, 2013 Feb;14(2):136-42), we now refer to those results instead of republishing the repetitive EM data.

Specifically, we deleted Figure 5C and instead state that ‘These results were in the full agreement with the previous electron microscopy experiments that revealed extensive damage and vacuolization of the mitochondria in Paneth cells during *T. gondii* infection

– How does IFN γ induced cell death compares to the cell death triggered by other cytokines, such as TNF?

This is both a very interesting and independent question. As suggested by all reviewers and editors, we chose to tone down the language since only a subset of the experiments were performed. In this work we have addressed the question of how IFN-γ ‘kills’ Paneth cells. We were able to reveal that this is a direct and mTOR-dependent mechanism. It is not trivial to compare it with the TNF-mediated mechanism as while this pathway is much better studied in general, the correct experiments would require development of additional in vivo models to study the effects of TNF on Paneth cells only (Paneth cell specific TNFRI KO mice as well as PC-specific TNFRII KO mice).

B) Regarding the impact of the microbiota on mTORC and Paneth cells:The authors also show that intestinal microbiota contributes to mTOR dysfunction and Paneth cell death. However the data shown in Figure S6D is not called out in the results and this should be corrected. Is the reason microbiota depletion rescues mTOR activity because the bacteria are necessary for IFN-g production during T. gondii infection? Please clarify this point in the manuscript. It should be noted that degranulation of Paneth cells is not stimulated in germ-free mice. Thus, Paneth cells might be more resilient to mTOR inhibition in germ-free mice. A more quantitative approach for the assessment of mTOR signaling in SPF versus WT mice across time, including earlier time points than 7 days post infection (see Figure S6) and additional markers of mTOR signaling is needed.

We apologize for not being clear in our message. We only claim that the reduced levels of IFN-γ seen in GF mice (now Figure 7 – supplement 1) are responsible for the survival of Paneth cells in the absence of microbiota (Figure 7). These results are consistent with our previous publications (Cell Host Microbe. 2009; 6 187-196) in which we revealed that microbiota boosts IFN-γ during *T. gondii* infection.

We have now performed an additional experiment of injection of rIFN-γ in GF mice. We observed that GF mice treated with rIFN-γ loose PC similarly to CV mice (Figure 7 – supplement 1C). We hope that this addressed the reviewer’s concern and we again apologize for not delivering this simple message in our original submission.